# Overexpression of *RCAN1*, a Gene on Human Chromosome *21*, Alters Cell Redox and Mitochondrial Function in Enamel Cells

**DOI:** 10.3390/cells11223576

**Published:** 2022-11-11

**Authors:** Yi Li, Veronica Costiniti, Guilherme H. Souza Bomfim, Maria Neginskaya, Ga-Yeon Son, Beverly Rothermel, Evgeny Pavlov, Rodrigo S. Lacruz

**Affiliations:** 1Department of Molecular Pathobiology, New York University College of Dentistry, New York, NY 10010, USA; 2Department of Internal Medicine and Molecular Biology, UT Southwestern Medical Center, Dallas, TX 75390, USA

**Keywords:** ameloblasts, Down syndrome, mitochondria, redox

## Abstract

The regulator of calcineurin (RCAN1) has been implicated in the pathogenesis of Down syndrome (DS). Individuals with DS show dental abnormalities for unknown reasons, and RCAN1 levels have been found to be elevated in several tissues of DS patients. A previous microarray analysis comparing cells of the two main formative stages of dental enamel, secretory and maturation, showed a significant increase in RCAN1 expression in the latter. Because the function of RCAN1 during enamel formation is unknown, there is no mechanistic evidence linking RCAN1 with the dental anomalies in individuals with DS. We investigated the role of RCAN1 in enamel by overexpressing RCAN1 in the ameloblast cell line LS8 (LS8^+RCAN1^). We first confirmed that RCAN1 is highly expressed in maturation stage ameloblasts by qRT-PCR and used immunofluorescence to show its localization in enamel-forming ameloblasts. We then analyzed cell redox and mitochondrial bioenergetics in LS8^+RCAN1^ cells because RCAN1 is known to impact these processes. We show that LS8^+RCAN1^ cells have increased reactive oxygen species (ROS) and decreased mitochondrial bioenergetics without changes in the expression of the complexes of the electron transport chain, or in NADH levels. However, LS8^+RCAN1^ cells showed elevated mitochondrial Ca^2+^ uptake and decreased expression of several enamel genes essential for enamel formation. These results provide insight into the role of RCAN1 in enamel and suggest that increased RCAN1 levels in the ameloblasts of individuals with DS may impact enamel formation by altering both the redox environment and mitochondrial function, as well as decreasing the expression of enamel-specific genes.

## 1. Introduction

Enamel, the outer layer of the tooth, is formed by differentiated epithelial cells called ameloblasts in two main functional stages: secretory and maturation [1,2]. In the secretory stage, matrix proteins are secreted into the enamel space with full enamel mineralization taking place during the maturation stage [1]. The organic matrix of enamel is composed of several proteins unique to this environment, such as ameloblastin, amelogenin, and enamelin, along with a subset of proteases [3]. Our previous microarray analysis of secretory and maturation stage ameloblasts to identify molecular requirements driving enamel maturation revealed a significant increase in the expression of the regulator of calcineurin 1 (RCAN1) in the maturation stage [2]; however, its function in enamel formation is unknown.

RCAN1 (also called Down syndrome candidate region 1 (DSCR1)) is located on human chromosome 21 (HSA21) and regulates the activity of calcineurin, a calcium/calmodulin-dependent protein phosphatase. As a consequence, RCAN1 impacts cellular signaling through calcineurin-dependent transcript factors, such as NFAT (nuclear factor of activated T cells) [4]; plays a role in mitochondrial function [5]; and has been implicated in several Down syndrome (DS) pathologies. RCAN1 expression is elevated in several tissues from DS patients (e.g., heart, brain) [6]. DS is caused by trisomy of all or part of human chromosome 21 (HSA21) and is the most common genetic cause of intellectual disability affecting 1 in 700 births worldwide [7]. DS patients show a high incidence of enamel defects, including malformed teeth, delayed tooth eruption, and hypoplasia [8,9,10]. However, the molecular bases causing these disruptions in the enamel of individuals with DS are unknown.

Our recent studies demonstrated that mitochondria participate in the maintenance of Ca^2+^ homeostasis in ameloblasts [11] and that there is an increase in reactive oxygen species (ROS) levels in maturation-stage ameloblasts [12]. Because overexpression of RCAN1 can cause mitochondrial dysfunction and increase the susceptibility of cells to oxidative stress [13], we postulated that ameloblasts of DS patients may experience a disruption in mitochondrial function, or enhanced ROS, altering enamel formation.

To address the role of RCAN1 in enamel development, we overexpressed *Rcan1* in the enamel cell line LS8, which models the secretory stage [14]. We show that both mitochondrial metabolic function and Ca^2+^ handling were affected by *Rcan1* overexpression and the expression of several enamel-specific genes. These data suggest that RCAN1 could be an important modulator of mitochondrial function and enamel gene expression during development and, as a consequence, contribute to alterations in enamel formation in individuals with DS.

## 2. Materials and Methods

### 2.1. Animal Use

All procedures employed in this study were conducted in accordance with guidelines approved by the Institutional Animal Care and Use Committee (IACUC) of the New York University College of Dentistry (IA16-00625).

### 2.2. Overexpression of RCAN1 in LS8 Cells

The RCAN1 expression plasmid (Sino Biological Inc., Houston, TX, USA #MG51377) or control (empty vector) plasmid were transfected into LS8 cells using FuGENE transfection reagent (Promega) following the manufacturer’s specifications. The ameloblast LS8 cells are a murine-derived immortalized cell line [15], a kind gift from Dr. M. L. Snead, and were maintained as reported [15]. Briefly, cells were cultured in DMEM with 10% fetal bovine serum (Gibco), 1% penicillin/streptomycin (Gibco), and 1% glutamine (Gibco) in a 5% CO_2_ incubator at 37 °C [1] and used at low passage.

### 2.3. Western Blot

Total protein was extracted from cell lysates prepared in an ice-cold RIPA (1% NP40, 0.1% SDS, 0.5% deoxycholate, 150 mM NaCl, 50 mM Tris pH 8.0) protease inhibitor cocktail (Complete Mini Protease Inhibitor Cocktail tablets from Roche). We homogenized the samples using a pestle prior to sonication, which we then cleared by centrifuging at 16,000 rpm for 15 min at 48 °C. Proteins were quantitated using Micro BCA Protein Assays (Pierce) and equally loaded at 15 mg/lane and resolved by SDS–polyacrylamide gel electrophoresis (SDS–PAGE) on 10% gels. Results were quantitated by densitometry using ImageJ (v.1.53) using the gray scale mode. We used the anti-mouse RCAN1 antibody (Abcam # ab140131). The experiment was repeated twice.

### 2.4. Immunofluorescence

Dental tissues were obtained for immunohistochemical analysis by ventricular perfusion of 4% PFA. Fixed hemimandibles were decalcified (10% EDTA, 2 weeks), embedded in paraffin, and sectioned (5 μm thick). Immunofluorescence was performed as reported [16] using the anti-DSCR1 antibody (D6694, SIGMA) using the biotin-labeled anti-rabbit IgG (1:500 dilution, Vector Laboratories), and detection was carried out using Streptavidin Alexa Fluor 488 (1:800 dilution, Life Technologies). Samples were embedded using Fluoromount mounting medium (Novus) containing DAPI (Thermo Fisher Scientific). Images were taken using a Leica TCS SP5 II confocal microscope.

### 2.5. RNA Extraction and Quantitative PCR

Total RNA was isolated using the RNeasy Micro Kit (Qiagen), as indicated by the manufacturer, followed by reverse transcription using the iScript cDNA Synthesis Kit (Bio-Rad Laboratories) and as reported elsewhere [16]. The primers used are listed in Appendix A. Samples were normalized to *GAPDH* or *HRPT*.

### 2.6. GSH/GSSG Assay

The GSH/GSSG was quantified as we have reported [12,17] using a luciferase-based kit (GSH/GSSG-Glo Assay, Promega), following the manufacturer’s instruction. The luminescence signal was recorded with a FlexStation 3 plate reader (Molecular Devices).

### 2.7. Mitochondrial Superoxide and OCR Measurements

LS8 cells were plated on 96-well plates and transfected. After 48 h, cells were washed and kept in a Ringer solution containing 5 mM pyruvate and 2 mM Ca^2+^ for 2 h at 37 °C. MitoSOX Red (5 μM) (Invitrogen) and 5 μM Cell Trace Violet (Invitrogen) were loaded (30 min, room temperature) [12]. After washing with the Ringer solution, fluorescence (450/480 nm) was measured in a FlexStation 3 plate reader (Molecular Devices). The Mitochondrial Stress Test Kit (Agilent) was used to analyze mitochondrial oxygen consumption in LS8 cells, as described [12,17,18]. Briefly, RCAN1-transfected LS8 cells and controls (empty vector) were seeded for 24 h in an XFe24-well microplate (Agilent) at 2500 cells per well in complete DMEM (10% FBS, 1% penicillin/streptomycin, and 1% glutamine). Oligomycin (1 μM), FCCP (1 μM), and rotenone/antimycin A (0.5 μM) (Agilent) were serially added in a Seahorse XFe24 analyzer. Cell plate and compound plate were loaded into a Seahorse XFe analyzer, and OCR was analyzed. After the run, the protein content of each well was analyzed by bicinchoninic acid (BCA), and data were normalized before analyzing basal respiration, ATP production, maximal respiration, and respiratory reserve.

### 2.8. Mitochondrial NADH Autofluorescence Measurements

NAD(P)H autofluorescence was measured as described previously [19]. Briefly, autofluorescence of mitochondrial NAD(P)H was excited using light from a Xenon lamp (Sutter Instrument Lambda Lb-ls/30) with λ = 380 nm. Autofluorescence (emission at 460 nm) was captured using a Nikon Eclipse TE2000-E fluorescent microscope equipped with a CCD camera. Images were acquired every 15 sec and analyzed using the ImageJ software. After the measurement of the basal level of NAD(P)H autofluorescence (~2 min), FCCP (1 µM) was added to maximize the oxidation of NADH, followed by the addition of NaCN (1 mM) to inhibit electron transport chain and achieve maximal reduction of NAD^+^. The basal NADH redox state was calculated after the signal was normalized as follows: maximal oxidation signal was set as 0%, and maximal reduction signal as 100%.

### 2.9. Mitochondrial Ca^2+^ Uptake

Cells were plated on 25 mm optical borosilicate poly-l-lysine–coated sterile glass covers (Sigma) at 80% confluence and loaded with rhodamine-2AM (Invitrogen) (2 μM, 20 min, at room temperature). The regular Ringer’s solution contained the following composition (in mM): 155.0 NaCl; 4.5 KCl; 2.0 CaCl_2_; 1.0 MgCl_2_; 10 D-glucose; and 10 HEPES, pH 7.4 adjusted with NaOH. In the nominally Ca^2+^-free Ringer’s solution, the CaCl_2_ was omitted, and the osmolarity maintained. Cells were perfused with ATP (100 μM) (Abcam) in a nominal Ca^2+^ solution 60 sec after starting the recordings, and then both ATP and Ca^2+^ were removed 120 s after starting the recordings. Fluorescence recordings were obtained using a Nikon Ti2-E Eclipse inverted light microscope equipped with a digital SLR camera controlled by a computer software (NIS-Elements, USA). Rhodamine-2AM fluorescence intensity (555 nm), background corrected, was measured per region of interest. Fluorescence images were generated at 5 sec intervals and normalized, and the values were calculated using Image J (1.53J).

### 2.10. Statistical Analysis

All data, mathematical analyses, and graphs were analyzed and/or generated using the GraphPad Prism software, version 9.3.1 (San Diego, CA, USA). The Ca^2+^ uptake rate (slope) was calculated in each individual trace and fitted by the nonlinear regression curve. The equation of this model has the following functions: Y = IF(X < X0, Y0, Y0 + (Plateau-Y0) ∗ (1 − exp(-K ∗ (X-X0)))). Differences between the means of the group data ± SEM, that fit a normal distribution, were analyzed using two-tailed unpaired Student’s t-test. The limit of significance was established, * *p* < 0.05; ** *p* < 0.01; *** *p* < 0.001; **** *p* < 0.0001, and n.s., nonsignificant, when compared with its respective control group. Each experiment was performed at least three times.

## 3. Results

### 3.1. RCAN1 Is Highly Expressed in Rat Maturation Stage Ameloblasts

We analyzed *Rcan1* expression in primary ameloblasts from the rat in the secretory and maturation stages by qRT-PCR. We show that *Rcan1* expression is significantly higher in maturation-stage ameloblasts compared with secretory-stage cells (Figure 1A), confirming the array data [2]. Isoform-specific primers of the two RCAN1 isoforms (RCAN1.1 or RCAN1.4) showed that *Rcan1.4* is the more abundant isoform in rat ameloblasts (Figure 1B). *Rcan1.1* is constitutively expressed in many tissues, whereas *Rcan1.4* expression is increased in response to the activation of calcineurin [20], an abundant protein in rat maturation ameloblasts [21]. RCAN1 localization was assessed by immunofluorescence (Figure 1C), showing strong signals in maturation-stage (upper panel) ameloblasts with a weaker staining in the secretory ameloblasts (lower panel). These data confirm an increase in *Rcan1* expression in maturation-stage ameloblasts.

### 3.2. RCAN1 Overexpression in LS8 Cells Alters Redox and Mitochondrial Function

We overexpressed *Rcan1* in the ameloblast cell line LS8, a widely used model, to investigate amelogenesis that most closely simulates the secretory stage [14]. Cells were transiently transfected with either an *Rcan1* expression plasmid or an empty vector control and analyzed 48 h later for changes in RCAN1 protein levels by Western blot. *Rcan1* transfected LS8 cells (LS8^+RCAN1^) showed a robust increase in RCAN1 protein levels (Figure 2A,B). Because elevated levels of RCAN1 have been associated with changes in mitochondrial function [5,22], we analyzed whether this was also the case in LS8^+RCAN1^ cells. We analyzed redox changes by measuring the ratio of reduced (GSH) to oxidized (GSSG) glutathione, an indicator of the redox status in cells [23]. LS8^+RCAN1^ cells showed a significant decrease in the GSH/GSSG ratio indicative of increased oxidation (Figure 3A). Mitochondrial function was determined more directly by a Seahorse assay to measure the rate of oxygen consumption by mitochondria. LS8^+RCAN1^ cells showed lower basal respiration and ATP production than control cells (Figure 3C,D). However, this decrease in mitochondrial respiration was not attributed to changes in the expression of the mitochondrial complexes, complexes I–V of the ETC, as a Western blot analysis of these complexes did not reveal any changes in protein expression (Figure 3E). Because mitochondrial bioenergetics were decreased in LS8^+RCAN1^ cells without a change in the abundance of mitochondrial complexes, we considered whether RCAN1 overexpression may have changed the levels of electron donors (e.g., NADH). We measured NADH autofluorescence and showed that there were no differences in NADH levels between LS8^+RCAN1^ cells and controls (Figure 3F–H).

### 3.3. RCAN1 Overexpression Elevated Mitochondrial Ca^2+^ Uptake

We then investigated whether Ca^2+^ signaling was a possible contributor to changes in mitochondrial function in the LS8^+RCAN1^ cells because the activity of several Ca^2+^-dependent tricarboxylic acid (TCA) cycle enzymes is enhanced by mitochondrial Ca^2+^ [24]. We measured Ca^2+^ uptake in the mitochondria of LS8^+RCAN1^ cells and controls after ATP (100 μM) stimulation in the presence of external Ca^2+^ in the solution to induce IP_3_R-mediated Ca^2+^ release from the endoplasmic reticulum (ER). Enamel cells express all three IP_3_R subtypes [25], and they respond to stimulation by ATP [18]. ATP stimulation of LS8^+RCAN1^ cells loaded with the mitochondrial Ca^2+^ indicator Rhod-2AM showed significantly higher mitochondrial Ca^2+^ uptake than controls (Figure 4A–C).

### 3.4. Enamel Gene Expression Is Reduced by Overexpression of Rcan1

We have previously shown that alterations in redox affects the expression of several enamel genes [12]. Amelogenin (AMELX) is the dominant product secreted by the ameloblasts, followed by enamelin (ENAM) and ameloblastin (AMBN) [3]. Two important secreted enzymes (MMP-20, KLK4) also contribute to enamel formation by cleaving AMELX [3]. Because *Rcan1* overexpression elevates ROS in LS8 cells, we asked whether the expression of these enamel genes was altered. This is important because in individuals with DS, RCAN1 levels are elevated and could be contributing to some of the dental enamel defects associated with DS. We analyzed gene expression by qRT-PCR and show that *Amelx*, *Ambn*, and *Klk4* are all significantly downregulated in LS8^+RCAN1^ cells (Figure 5).

## 4. Discussion

*Rcan1.4* was found to be highly expressed in maturation-stage ameloblasts by qRT-PCR, confirming our previous array data and determining which of the specific *Rcan1* isoforms is involved (Figure 1A). Because the expression of the *Rcan1.4* isoform is directly under the control of calcineurin [26], this suggests that calcineurin activity increases in the transition from the secretory stage to the maturation stage. This is certainly in agreement with our demonstration that Ca^2+^ transport significantly increases at that stage [2,27] and would be predicted to increase the potential for activation of Ca^2+^-dependent signaling, such as those mediated through calcineurin. Although RCAN1′s function as a calcineurin inhibitor is well documented, its role in amelogenesis is unknown. We previously hypothesized that the increase in RCAN1 levels in the maturation stage could be involved in attenuating Ca^2+^-induced cell stress [27]. However, calcineurin carries out a wide array of cellular functions, and thus, by extension, changes in the levels of its endogenous inhibitor, RCAN1, are also likely to affect a diversity of processes. Of particular relevance to the studies presented here is that elevated RCAN1 levels have been shown to alter mitochondrial function and increase ROS generation [22,28]. The *RCAN1* gene is located on human chromosome 21 (HSA21), trisomy of which causes Down syndrome. Our interest in RCAN1 stems in part from the observation that DS syndrome patients present with several dental anomalies, including hypoplasia, a developmental defect caused by disruptions in amelogenesis. DS patients show elevated RCAN1 levels in a diversity of tissues [6], and therefore, it is expected that RCAN1 levels would be higher than normal in both stages of ameloblasts in the setting of DS. These connections prompted us to investigate whether overexpressing RCAN1 in an in vitro ameloblast cell model affected mitochondrial function and Ca^2+^ signaling, which could therefore explain, at least in part, why DS patients show enamel defects.

We have previously shown that mitochondrial respiration and ROS levels are lower in secretory-stage ameloblasts than in maturation-stage cells [12]. Thus, the physiology of secretory ameloblasts may be more sensitive to an elevation in ROS levels which could affect cell function. To examine the response of secretory-stage ameloblasts to increased RCAN1 levels, we overexpressed *Rcan1* in the enamel cell line LS8 (LS8^+RCAN1^), a cell model representing the secretory stage [14]. ROS levels were significantly higher in LS8^+RCAN1^ cells than controls, and these cells also showed deficient mitochondrial respiration (Figure 3). Interestingly, this decrease in mitochondrial respiration occurred without changes in the expression of the mitochondrial complexes I–IV or in ATP synthase (Figure 3E), or in NADH/NAD redox status (Figure 3F–H).

RCAN1 overexpression has been previously shown to be sufficient to elevate ROS generation without changing the overall mitochondrial content [29], consistent with our findings reported here. However, the precise cause of the corresponding decrease in mitochondrial respiration is unclear. Mitochondria can uptake cytosolic Ca^2+^ following its elevation in response to the opening of Ca^2+^ channels either in the plasma membrane, such as ORAI1 [11,30], or in the ER, such as IP_3_R [31]. The latter source is considered a more efficient mechanism for stimulating mitochondrial function [32]. Free Ca^2+^ in the mitochondrial matrix stimulates the activity of several mitochondrial dehydrogenases. Thus, Ca^2+^ uptake into the matrix can increase NADH generation by the TCA cycle, thereby increasing electron transport, O_2_ consumption, and ATP generation [33,34]. However, high levels of _m_[Ca^2+^] can also have detrimental effects on mitochondrial function [35] due to the inhibition of complex I by Ca^2+^ phosphate granules localized to the matrix. We measured _m_[Ca^2+^] uptake in LS8^+RCAN1^ cells and controls and found that the cells overexpressing *Rcan1* displayed higher _m_Ca^2+^ uptake (Figure 4A). Sullivan and coworkers [36] have shown that mitochondrial respiration can be inhibited by Ca^2+^ in a dose-dependent manner without altering NADH levels, similar to the effects reported here. Taken together, this suggests that the reduction in mitochondrial respiration we observed in LS8^+RCAN1^ cells could have been the result of elevated _m_Ca^2+^ uptake. In support of this notion is the observation that, although mitochondria of secretory- and maturation-stage ameloblasts show similar levels of _m_Ca^2+^ uptake in response to ATP stimulation [11], _m_Ca^2+^ uptake was higher in LS8^+RCAN1^ cells compared with controls. LS8 cells are most similar to secretory ameloblasts, which we postulate may be more sensitive to changes in ROS levels; however, increased RCAN1 levels may increase the capacity for _m_Ca^2+^ uptake in all stages of ameloblast development, with the metabolic outcome depending upon the physiological state and cell type in question.

We also analyzed whether RCAN1 overexpression affected the expression of enamel-specific genes that have been shown to be sensitive to ROS [12]. Our analysis showed that the expressions of *Amelx* and *Ambn,* as well as the enamel proteases *Mmp20* and *Klk4*, were all significantly downregulated. These data suggest that RCAN1 not only alters cellular functions in enamel cells, but attenuates the activity of key enamel genes.

The study presented here used an immortalized enamel cell line to overexpress RCAN1 to mimic the possible effects that would be expected in enamel cells of Down syndrome patients. However, whether this is the case in DS patients is unknown, and additional studies using mouse models of DS would be important to address this concept. In summary, our study shows that RCAN1 is highly upregulated in maturation-stage ameloblasts, and that it may function as a negative regulator of enamel gene expression. Overexpressing *Rcan1* in LS8 cells increases ROS generation and negatively affects mitochondrial bioenergetics. The enhancement in _m_Ca^2+^ uptake in these cells may have contributed to the decline in bioenergetics. Combined, these data provide an insight into the potential molecular basis for enamel defects in DS.

## Figures and Tables

**Figure 1 cells-11-03576-f001:**
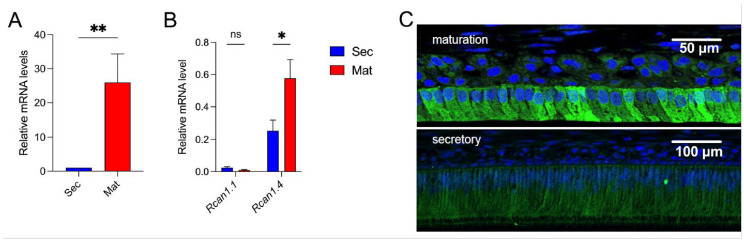
(**A**) Relative mRNA expression of *Rcan1* in rat secretory- and maturation-stage ameloblasts. (**B**) Relative mRNA expression of the two RCAN1 isoforms (*Rcan1.1*, *Rcan1.4*) in rat secretory- and maturation-stage ameloblasts. (**C**) Immunofluorescent localization of Rcan1 (green) in maturation- and secretory-stage ameloblasts. Dapi = blue. Data represent the mean ± SEM of three independent experiments analyzed by Student’s *t*-test. * *p* < 0.05, ** *p* < 0.01.

**Figure 2 cells-11-03576-f002:**
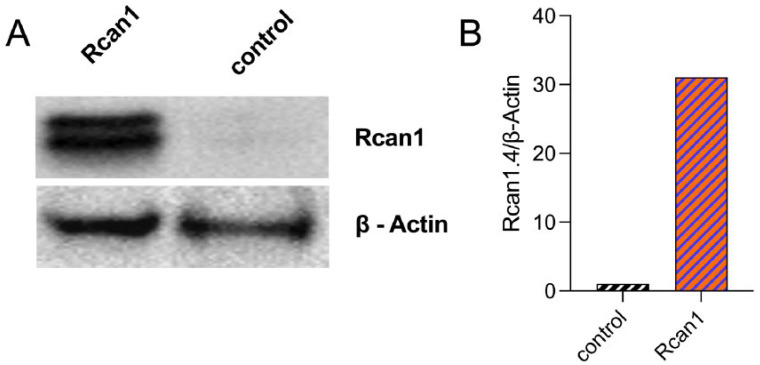
(**A**,**B**) Representative Western blot from two separate experiments showing overexpression of Rcan1 in LS8 cells, quantitated by ImageJ v.1.53j.

**Figure 3 cells-11-03576-f003:**
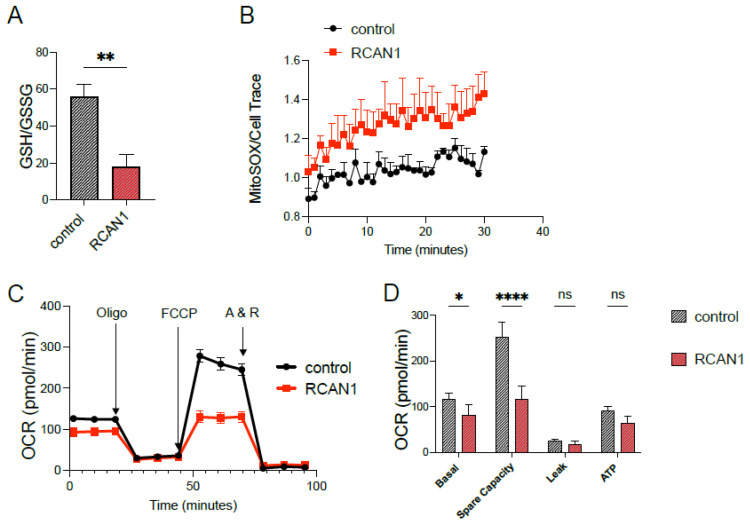
(**A**) Quantification of the reduced (GSH) to oxidized (GSSG) glutathione in LS8 overexpressing Rcan1. (**B**) Traces of MitoSOX measurements in control and Rcan1 overexpressing LS8 cells. (**C**) Oxygen consumption rate (OCR) measured in control and RCAN1 overexpressing LS8 cells. (**D**) Quantification of basal respiration, spare respiratory capacity, proton leak, and ATP-linked mitochondrial respiration. (**E**) Western blot showing the expression of the mitochondrial complex proteins I–IV and the ATP synthase in control and RCAN1 overexpressing LS8 cells. (**F**–**H**) NADH autofluorescence in control (**F**) and RCAN1 overexpressing (**G**) LS8 cells. (**H**) Quantification of the redox state. Data represent the mean ± SEM of three independent experiments analyzed by Student’s *t*-test. * *p* < 0.05, ** *p* < 0.01, **** *p* < 0.0001.

**Figure 4 cells-11-03576-f004:**
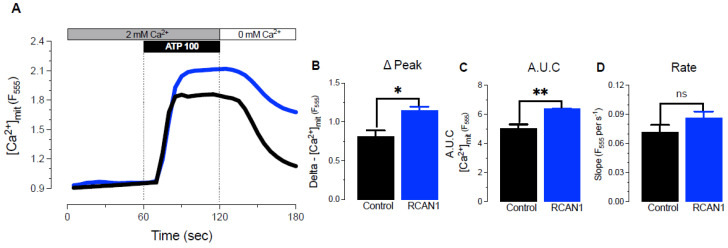
(**A**) Mitochondrial Ca^2+^ uptake in control and RCAN1 overexpressing LS8 cells using Rhod-2AM after ATP (100 μM) stimulation. Black traces = control, blue traces = RCAN1. (**B**–**D**) Quantification of the ∆ peak of Ca^2+^ influx, area under the curve (A.U.C), and Ca^2+^ influx rate in control (black) and RCAN1 overexpressing (blue) LS8 cells. Data represent the mean ± SEM of three independent experiments analyzed by Student’s *t*-test. * *p* < 0.05, ** *p* < 0.01.

**Figure 5 cells-11-03576-f005:**
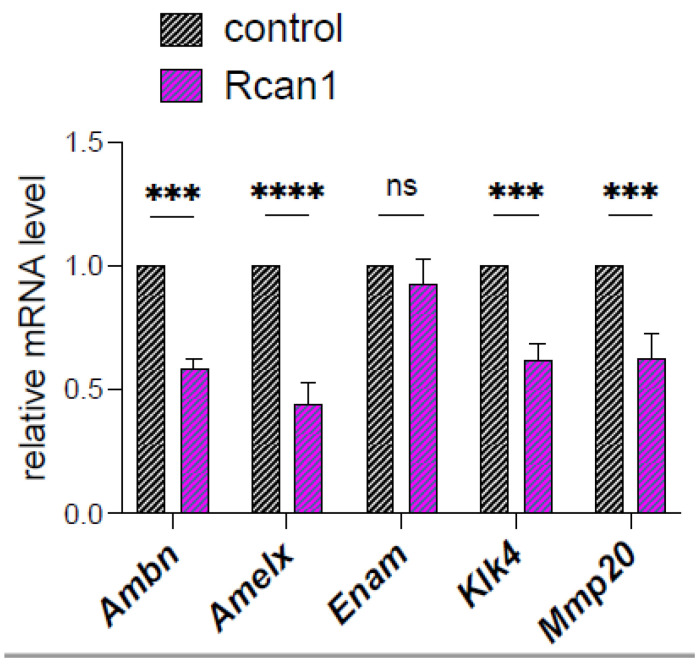
Relative mRNA expression by qRT-PCR of enamel genes in control and RCAN1 overexpressing LS8 cells. Data represent the mean ± SEM of three independent experiments analyzed by Student’s *t*-test. *** *p* < 0.001, **** *p* < 0.0001.

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
