# Peer review of "Overexpression of RCAN1, a Gene on Human Chromosome 21, Alters Cell Redox and Mitochondrial Function in Enamel Cells"

_cells, 2022, doi:10.3390/cells11223576_

Round 1

Reviewer 1 Report

Article title “Overexpression of RCAN1 gene linked to Down syndrome alters cell redox and mitochondrial function in enamel cells” by Li et al., describes RCAN1 gene linked to Down syndrome, and alters mitochondrial function.

The overall article is very interesting, and collection of information reported are useful in emphasizing critical findings, which directly supports authors theory. I have no concerns related to the literature cited and findings mentioned in this article. However, if possible, I would recommend another small experiment, as describes below.

Result 3.3: Author tested the hypothesis of increase in Ca2+ uptake by using an elevated Ca2+ solution. Is it possible to perform another experiment at low Ca2+ or in presence of internal Ca2+ chelators that will show that indeed the effect we are observing are mostly due to increases in external Ca2+ uptake.

Manuscript is well written.

Reviewer 2 Report

The manuscript investigates the role of Regulator of Calcineurin 1 (RCAN1) in enamel forming ameloblasts.  The study is interesting in giving insight into basic cellular mechanism of enamel formation, and because of the potential role of RCAN1 in Down’s syndrome dental pathology, RCAN1 having been previously shown to be elevated in tissues from DS patients.  The manuscript is clearly written, and the contains some interesting data.

The authors have a well worked out mechanistic explanation for their observations.  They first show differential expression of RCAN1 in maturation stage ameoblasts by RT-PCR and immunocytochemistry, consistent with previous studies.  They next overexpress RCAN1 in a secretory-type ameoblast cell line (LS8), which may shed light on consequences of RCAN1 overexpression in DS.  Overexpression of RCAN1 in LS8 cells causes an increase in ROS and a decrease in mitochondrial oxidative respiration (by Seahorse analyser).  Overexpression also causes decreased expression of several enamel related genes (Figure 4, qRT-PCR).  The authors have previously reported that increased ROS (via H2O2) can alter enamel gene expression (Li et al., 2021).  Finally, the authors attempt to link increased ROS to an observed elevation in mitochondrial Ca2+ (measured by Rhod-2) via a ROS-mediated increase in release of IP3-sensitive Ca2+ pools.  IP3-R subtypes are elevated (via qRT-PCR).  Although the data are broadly consistent with the authors interpretation, the mechanistic link to IP3 receptors seems a bit speculative.   Does provision of ROS (e.g. H2O2) elevate mitochondrial Ca2+?  Have other mechanisms (e.g. changes in mito Ca2+ handling, SOCE, Ca2+ ATPases) been considered/eliminated?  Presumably Rcan1 overexpression could have quite complicated effects on Ca2+ homeostasis.   

Minor points:

1.       Methods: Please give source for ab140131

2.       Methods: please state origin of LS8 cells (mouse derived?)

3.       Methods and Figure 2:  please give explain the rationale for the protocol used to assess mitochondrial Ca2+.  Is ATP added to trigger P2Y-mediated Ca2+ signals?  Why is extracellular Ca2+ switched form 2 mM to 0 mM?  Is this to assess mito Ca2+ elevation and extrusion separately?  As the peak in Rcan1 transfected cells is higher, one might have expected the rising phase to be steeper.  Do the authors have any explanation why this isn’t seen.  (What is the time course of solution exchange, this may be relevant here).  Also, mito Ca2+ decline seems rather slow, with a long plateau.  Is this expected? 

Reviewer 3 Report

1. In the abstract, a previous study has showed that the expression of RCAN1 is higher in the individuals with DS, which indicates a potential association between RCAN1 and DS. 

2. Why did the authors assess the mitochondrial functions after  overexpression of RCAN1? Related information should be added in the abstract. 

3. The processes of WB should be an independent part in the methods.

4. More information about WB should be emphasized in the methods, such as the concentration of primary antibody, the incubation duration and so on. 

5. I recommend the authors add the WB analysis of RCAN1 related to figure 1A and 1B. 

6. The figure 1D and 1E can be apart from figure 1 and added into figure 2.

7. In figure 1D, was there no expression of RCAN1 in the control group?

8. Why did the authors utilize LS8 cells to perform the further experiments, but not using the maturation stage ameloblasts? Although the transfection of RCAN1 plasmid increased the expression of RCAN1 about 30 times, the RCAN1 expression is approximately 20 times higher in maturation stage ameloblasts compared to secretory stage cells. 

9. In figure 1, significant label should be added in 1E.

10. The average Rcan1 expression is more likely to be 20 times higher in maturation stage ameloblasts compared to secretory stage cells from figure 1A.

11. In the part 3.4, were the expression of these genes changed in maturation stage ameloblasts compared to secretory stage cells?

12. WB is needed to analyze the enamel function, such as the expression of AMELX, ENAM and so on.

13. The discussion needs re-written, and the flow is puzzled. The authors should show results them according to previous study, and discuss some specific results, then exploring potential causes. 

14. Limitations and further research plans are needed in the discussion. 

15. The title is not appropriate, there is no results in the study indicating the overexpression of RCAN1 gene linked to Down syndrome alters cell redox.

Round 2

Reviewer 3 Report

1. If there was little expression of RCAN1 in LS8 cells, it might be a serious mistake to overexpress RCAN1 by using LS8 because RCAN1 is very likely to exhibit no function in LS8 under physiological circumstance. It may mean that all experiments by using LS8 cells should be re-performed.

2. Although the expression of AMELX, ENAM may be very low in LS8 cells, AMELX and ENAM severe  as proteins to exhibit  functions instead of mRNA. Thus, the WB analysis is necessary. Else, it was not suitable to regard these protein as markers. 

3. Overexpression of RCAN1 did not only indicate an association with Down synfrome, but may some other diseases. Moreover, in the manuscript, the authors did not use any samples from DS patients or animal model. In other words, The authors can emphasize the harm of RCAN1 overexpression in the abstract and introduction, sucha as "high expression of RCAN1 links to DS", but it is not approprate to state "Overexpression of RCAN1 gene linked to Down syndrome" in title becasue the information is derived from previous study rather than the results in this manuscript.